# Electronic Structures of Kitaev Magnet Candidates RuCl_3_ and RuI_3_

**DOI:** 10.3390/nano14010009

**Published:** 2023-12-19

**Authors:** Subhasis Samanta, Dukgeun Hong, Heung-Sik Kim

**Affiliations:** 1Department of Physics, Kangwon National University, Chuncheon 24341, Republic of Korea; samanta@kangwon.ac.kr (S.S.); dghong@kangwon.ac.kr (D.H.); 2Institute of Quantum Convergence and Technology, Kangwon National University, Chuncheon 24341, Republic of Korea

**Keywords:** Kitaev magnetism, spin–orbit coupling, layered compounds, honeycomb lattice, first-principles electronic structure calculation, density functional theory, dynamical mean-field theory

## Abstract

Layered honeycomb magnets with strong atomic spin–orbit coupling at transition metal sites have been intensively studied for the search of Kitaev magnetism and the resulting non-Abelian braiding statistics. α-RuCl${}_{3}$ has been the most promising candidate, and there have been several reports on the realization of sibling compounds α-RuBr${}_{3}$ and α-RuI${}_{3}$ with the same crystal structure. Here, we investigate correlated electronic structures of α-RuCl${}_{3}$ and α-RuI${}_{3}$ by employing first-principles dynamical mean-field theory. Our result provides a valuable insight into the discrepancy between experimental and theoretical reports on transport properties of α-RuI${}_{3}$, and suggests a potential realization of correlated flat bands with strong spin–orbit coupling and a quantum spin-Hall insulating phase in α-RuI${}_{3}$.

## 1. Introduction

Kitaev’s exactly solvable honeycomb lattice model, hosting Majorana quasiparticles and non-Abelian braiding statistics, has attracted enormous interest recently, due to the potential fault-tolerant topological quantum computations that it promises [1]. A subsequent theoretical suggestion by G. Jackeli and G. Khaliullin, the so-called Jackeli–Khaliullin mechanism [2,3], paved a direction towards the realization of Kitaev’s frustrated anisotropic exchange interactions in solid-state systems, which in an ideal situation should result in the Kitaev spin liquid phase. This initiated a new field of Kitaev magnetism study and intensive theoretical and experimental follow-up investigations [4,5,6,7,8,9].

Among the material candidates, α-RuCl${}_{3}$ has been considered the most promising candidate [10,11,12,13,14,15,16]. However, the nonvanishing zigzag-type antiferromagnetic order in the compound, albeit suppressed by external magnetic fields [14] has hindered true realization of the Kitaev spin liquid phase. In this regard, enhancing hybridizations between the Ru and halide ions by replacing Cl into Br and I in α-RuCl${}_{3}$ has been considered a viable path toward realizing ideal Kitaev exchange interactions and the resulting spin liquid state [17,18,19].

Fortunately, there have been several experimental reports on successful syntheses of α-RuBr${}_{3}$ [20,21] and α-RuI${}_{3}$ [22,23], focusing on the possibility of promoting and realizing the Kitaev spin liquid phase. Interestingly, α-RuI${}_{3}$ was reported to be metallic, but with exceptionally high resistivity [23]. On the other hand, a theoretical study employing the density functional theory + *U* (DFT + *U*) method presents a magnetic and insulating phase for α-RuI${}_{3}$ [24]. In this study, it was speculated that the bad-metallic state observed experimentally can be due to sample quality issues, and especially due to formations of metal grain boundaries between insulating RuI${}_{3}$ grains [24]. Later DFT + *U* studies provide a partial explanation of this discrepancy between experimental and theoretical observations by choosing a suitable *U*-value that yields insulating and metallic phases in α-RuCl${}_{3}$ and α-RuI${}_{3}$, respectively [25,26]. However, the observed high resistivity in α-RuI${}_{3}$, which goes beyond simple band descriptions, still raises questions about the true nature of the metallic character and potential effects of electron correlations in the compound [23].

To address these issues, we study electronic structures of α-RuCl${}_{3}$ and α-RuI${}_{3}$ by employing first-principles dynamical mean-field theory combined with density functional theory (DFT + DMFT) in a comparative manner. Specifically, we focused on the impact of dynamical electron correlations on the Mott-insulating and potentially correlated metallic phases of α-RuCl${}_{3}$ and α-RuI${}_{3}$, respectively, which cannot be captured within conventional DFT and DFT + *U* approaches. In α-RuCl${}_{3}$, we produce the paramagnetic Mott-insulating phase with the formation of the spin-orbit entangled Ru $j_{\mathrm{eff}}$ = 1/2 local moment [11,17,27,28]. On the other hand, in α-RuI${}_{3}$, we observe a metallic phase with strongly renormalized almost-flat bands consisting of the $j_{\mathrm{eff}}$ = 1/2 orbital character. Therefore, α-RuI${}_{3}$ can be considered a correlation-induced flat band system with strong spin–orbit coupling effects, where the flat bands are located exactly at the Fermi level and may give rise to the bad-metallic character, as observed experimentally, due to its heavy electron mass and other flat-band-induced instabilities [23]. We further suggest that exfoliating α-RuI${}_{3}$ may result in an insulating sheet of single-layer RuI${}_{3}$, which can be driven into Mott-insulating or topological quantum spin-Hall phases. This observation calls for further studies on the nature of the correlated flat bands in the presence of long-range Coulomb interactions and potential intriguing electronic instabilities in α-RuI${}_{3}$.

## 2. Computational Methods

A fully charge-self-consistent DMFT method [29], implemented in DFT + embedded DMFT (eDMFT) functional code [30] combined with the wien2k package [31], was employed for calculating electronic structures and relaxing internal atomic coordinates [32]. In the DFT part, Ceperley–Alder (CA) local density approximation (LDA) was employed [33], and 2000 *k*-points were used to sample the first Brillouin zone with $RK_{\max}$ = 7.0. A force criterion of 10${}^{-4}$ Ry/Bohr was adopted for optimizations of internal coordinates. A continuous-time quantum Monte Carlo method in the hybridization-expansion limit (CT-HYB) was used to solve the auxiliary quantum impurity problem [34], where the Ru $t_{2g}$ orbital was chosen as our correlated subspace in a single-site DMFT approximation. For the CT-HYB calculations, up to 1.5 × 10${}^{9}$ Monte Carlo steps (at *T* = 232 K) were employed for each Monte Carlo run. We checked that lowering *T* down to 58 K in the Monte Carlo runs did not affect qualitatively the nature of our results.

The reasonable hybridization window of −10 to +10 eV (with respect to the Fermi level) was chosen, and *U* = 6∼10 eV and $J_{H}$ = 0.8 eV of on-site Coulomb repulsion and Hund’s coupling parameters were used for the Ir $t_{2g}$ orbitals. Note that the *U*-value employed in eDMFT calculations should be larger than that used in DFT + U studies, due to differences in consideration of electron screening processes between eDMFT and DFT + U methodologies [35,36]. Also note that the *U*-value used in this study is higher than the value employed in other eDMFT studies on iridate compounds [37,38,39], *U* = 4.5∼5.0 eV, which is acceptable considering Ru 4*d* orbitals are more localized than the Ir 5*d* ones. Discussion on the choice of the *U*-value and the effect of *U*-tuning will be discussed below.

Note that, in our calculations, we fully incorporated atomic spin–orbit coupling within the Ru $t_{2g}$ orbitals. Inclusion of the spin–orbit coupling transforms the six (three orbitals × two spin components) orbitals into the so-called $j_{\mathrm{eff}}$ = 1/2 and 3/2 orbitals, as follows [27,28];
$$\begin{array}{cc} |j_{\mathrm{eff}}=\frac{1}{2};\pm \frac{1}{2}\rangle & =\mp \frac{1}{\sqrt{3}}\left(\right|d_{xy},↑↓\rangle \pm |d_{yz},↓↑\rangle +i|d_{xz},↓↑\rangle )\\ |j_{\mathrm{eff}}=\frac{3}{2};\pm \frac{1}{2}\rangle & =\sqrt{\frac{2}{3}}\left(|d_{xy},↑↓\rangle \mp \frac{|d_{yz},↓↑\rangle \pm i|d_{xz},↓↑\rangle }{2}\right)\\ |j_{\mathrm{eff}}=\frac{3}{2};\pm \frac{3}{2}\rangle & =\mp \frac{1}{\sqrt{2}}\left(\right|d_{yz},↑↓\rangle \pm i|d_{xz},↑↓\rangle ,) \end{array}$$
which are characterized by the effective total angular momentum quantum numbers $j_{\mathrm{eff}}$ and $j_{\mathrm{eff}}^{z}$. Note that the Ru $t_{2g}$ shell behaves as effective orbital angular momentum eigenstates, with $l_{\mathrm{eff}}$ = 1 ($|l_{\mathrm{eff}}=1;l_{\mathrm{eff}}^{z}=0\rangle \equiv |d_{xy}\rangle$, $|l_{\mathrm{eff}}=1;l_{\mathrm{eff}}^{z}=\pm 1\rangle \equiv \mp (|d_{yz}\rangle \pm i|d_{xz}\rangle )/\sqrt{2}$). Here, by “effective” we mean that the $t_{2g}$ orbitals are not exactly the l=1 orbital momentum states, and that we obtain an additional minus sign in the spin–orbit coupling term ($l\cdot s\to -l_{\mathrm{eff}}\cdot s$). Combined with spin s=1/2 of the electron, this $\left[l_{\mathrm{eff}}=1\right]⊗\left[s=1/2\right]$ complex splits into a $j_{\mathrm{eff}}$ = 1/2 doublet and 3/2 quadruplet. These $j_{\mathrm{eff}}$ orbitals become convenient bases for the electronic structure description and are chosen for the orbital projections in the density of states plots. Note also that, to reduce the sign problems in the Monte Carlo calculations, an Ising-type (density–density) Coulomb interaction was chosen.

## 3. Results

### 3.1. Comparison between α-RuCl${}_{3}$ and α-RuI${}_{3}$

Figure 1 shows the crystal structures of α-RuCl${}_{3}$ and α-RuI${}_{3}$ in the rhombohedral $R\bar{3}$ space group symmetry. We employed lattice parameters and internal coordinates from previous experimental studies [22,40], after which internal coordinates were optimized within our DFT + DMFT calculations. The differences between the experimental and DMFT-optimized internal atomic coordinates are less than 0.03 Å and are not shown in this work.

Figure 2a,b show quasiparticle spectral functions of α-RuCl${}_{3}$ and α-RuI${}_{3}$, obtained from DFT + DMFT calculations, respectively. Left panels in Figure 2a,b show false-color maps of momentum-dependent spectral functions A(k,ω), corresponding to band structures from conventional DFT calculations, with the blurring induced by quasiparticle scattering effects by self-energies [41,42]. Right panels show momentum-integrated and orbital-projected spectral functions, corresponding to projected density of states (PDOS) from DFT calculations. For this plot, on-site Coulomb repulsion and Hund’s coupling parameters for the quantum impurity problems were chosen to be 6 and 0.8 eV, respectively.

In both systems, Ru $e_{g}$ bands are well separated from Ru $t_{2g}$ states by about 2 eV, with little mixture between $t_{2g}$ and $e_{g}$ characters near the Fermi level, justifying our choice of Ru $t_{2g}$ as the correlated subspace for the impurity problem. It is also noticeable that the splitting between $e_{g}$ bands in α-RuI${}_{3}$ (Figure 2b) is larger than in α-RuCl${}_{3}$ (Figure 2a), which signals larger crystal field effects in α-RuI${}_{3}$ due to the enhanced hybridization.

From Figure 2a, a Mott-insulating gap of about 1.8 eV can be seen in α-RuCl${}_{3}$. This gap value is consistent with a previous experimental observation in the compound [43], justifying our choice of *U* and $J_{H}$ values. In addition, an almost pure $j_{\mathrm{eff}}$ = 1/2 (red curve in the right panel of Figure 2a) orbital character can be seen from the upper Hubbard band (around 1 eV above the Fermi level), signifying the presence of the spin–orbit-entangled $j_{\mathrm{eff}}$ = 1/2 local moment in α-RuCl${}_{3}$, originating from the cooperation of the Ru spin–orbit coupling and on-site Coulomb interactions, as previously reported [11,17].

On the other hand, Figure 2b shows a metallic electronic structure of α-RuI${}_{3}$. This metallic behavior has been reported previously and attributed to the larger hybridization between the Ru 4*d* and I 5*p* orbitals in α-RuI${}_{3}$ than that between the Ru 4*d* and Cl 3*p* orbitals in α-RuCl${}_{3}$ [19,25,26]. A larger I 5*p* orbital character, in addition to strong mixing between the Ru $j_{\mathrm{eff}}$ = 1/2 and 3/2 characters, can be seen from the right panel of Figure 2b, in a consistent manner with previous theoretical results (schematically illustrated in Figure 2c,d) [19,25,26]. Also note that the out-of-plane band dispersion (between the Γ and A points) of the $j_{\mathrm{eff}}$ = 1/2 bands at the Fermi level is not significant, manifesting the quasi-two-dimensional nature of the $j_{\mathrm{eff}}$ = 1/2 bands despite the large interlayer I-I hybridizations in this system.

### 3.2. Robust Metallic Character against the On-Site Coulomb Repulsion in α-RuI${}_{3}$

It is notable that the bandwidth of the $j_{\mathrm{eff}}$ = 1/2-like bands close to the Fermi level in α-RuI${}_{3}$ is about 0.25 eV (see Figure 2b), suppressed by about 50% compared with previous nonmagnetic DFT + *U* results [19,25,26]. This bandwidth renormalization is due to the dynamical correlation effects inherent in DMFT calculations. A natural question to follow is how α-RuI${}_{3}$ is close to the phase boundary between the metallic and insulating phases, or, equivalently, whether the metallic phase remains stable or becomes insulating as the on-site Coulomb parameter, *U*, is increased or the system reaches a two-dimensional limit.

To answer this question, we performed calculations with enhanced *U*-values. Figure 3 presents the results, where Figure 3a,b shows spectral functions with *U* = 8 and 10 eV, respectively ($J_{H}$ = 0.8 eV in both cases). As *U* is enhanced (see Figure 3a,b), the bandwidth renormalization and the eventual Mott-insulating phase at *U* = 10 eV is observed. Note, however, that *U* = 10 eV is an unacceptably large value for the Ru $t_{2g}$ orbital, and that *U* = 6 eV reasonably reproduces the size of the single-particle gap from photoemission and inverse photoemission results in α-RuCl${}_{3}$ [43]. Hence, we speculate that the correlated metallic phase remains stable in α-RuI${}_{3}$. Considering that almost flat bands in the vicinity of the Fermi level may be prone to various instabilities, this observation might be the origin of the sample dependence in the material properties of α-RuI${}_{3}$, as reported previously, where the presence of impurities or grain boundaries may lead to domains of distinct ground states [22,23,24]. Also, the correlation-induced band flattening and quasiparticle scattering may give rise to the bad-metallic character, as observed experimentally [23].

### 3.3. Potential Quantum Spin-Hall Insulator in the Single-Layer α-RuI${}_{3}$

In a previous DFT + *U* study, it was suggested that exfoliating the system and realizing the single-layer limit may drive the system into the insulting regime [26]. To check this, we performed a DMFT calculation of the single-layer RuI${}_{3}$ with the relaxation of internal atomic coordinates. Figure 4a shows the result, with the choice of (*U*, $J_{H}$) = (6, 0.8) eV. Interestingly, a clear pseudogap feature is observed. By plotting quasiparticle band dispersion by computing spectral function with the imaginary part of the self-energy set to be 0, depicted as white dotted lines in Figure 4a, a clear band gap of about 40 meV is observed.

The band-like character of the $j_{\mathrm{eff}}$ = 1/2 can be checked from the self-energy ${\hat{\Sigma }}_{\sigma }\left(E\right)$. In DMFT calculations, the spectral function can be computed from the single-particle band dispersion and the self-energy as follows;
(1)A^(k,E)=−1πImG^k(E),
where
(2)$${\hat{G}}_{k}\left(E\right)={\left(E-\mu +{\hat{H}}_{k}-\hat{\Sigma }\left(E\right)\right)}^{-1}.$$
Here, ${\hat{G}}_{k}\left(E\right)$ and $\hat{A}\left(k,E\right)$ are the Green’s function and the spectral function, while ${\hat{H}}_{k}$ and $\hat{\Sigma }\left(E\right)$ are the single-particle band Hamiltonian from DFT calculations and the self-energy from the many-body quantum impurity problem, respectively [41]. The hat and boldface used for $\hat{A}\left(k,E\right)$, ${\hat{G}}_{k}\left(E\right)$, ${\hat{H}}_{k}$, and $\hat{\Sigma }\left(E\right)$ denote that these symbols are represented as matrices with spin–orbital indices. Note that the Mott-insulating phase is characterized by the presence of peaks in ImΣ^(E) close to the Fermi level, which demonstrates quasiparticle scatterings at the atomic sites from the Coulomb repulsion [41].

The rightmost panel in Figure 4a shows that both the $j_{\mathrm{eff}}$ = 1/2 and 3/2 states show almost vanishing −ImΣ^(E) for both states close to the Fermi level. This shows that the effect of Coulomb repulsion, which introduces quasiparticle scatterings and the resulting Mott-insulating behavior, is marginal at *U* = 6 eV, even in the single-layer limit. Considering that the presence of $j_{\mathrm{eff}}$ = 1/2 orbitals hosts nontrivial complex second nearest neighbor hopping integrals, hence realizing Kane–Mele model-like electronic structures [44,45], this phase can be considered as a candidate of the quantum spin-Hall-like effect. Note that a similar suggestion was made on a potential realization of the quantum anomalous Hall phase in a fictitious ferromagnetic RuI${}_{3}$ single layer [46].

A direct confirmation of the topological nature of this phase can be tricky, because of the presence of electron correlations that blur the band description. Hence, we made an indirect check by constructing Wannier functions of the four $j_{\mathrm{eff}}$ = 1/2 quasiparticle band dispersions (i.e., bands computed with $\hat{\Sigma }\left(E\right)=0$, depicted as white dotted lines in Figure 4a) via employing the wien2wannier package [47]. To check the topological character, parity eigenvalues of the unoccupied $j_{\mathrm{eff}}$ = 1/2 bands at four time-reversal-invariant momenta (i.e., Γ and three M-points) were obtained from the Wannier-constructed $j_{\mathrm{eff}}$ = 1/2 tight-binding model [48]. The result shows that the band-like insulating phase of the single-layer RuI${}_{3}$ at *U* = 6 eV, shown in Figure 4a, is topologically trivial. Note that it can be driven into the quantum spin-Hall regime by applying an in-plane uniaxial strain, which induces band inversion at one of the three M-points depending on the direction of the strain [49].

Next, we check our calculation results for higher *U*-values. Figure 4b show the spectral functions and −ImΣ^(E) at *U* = 8 eV. From the left and middle panels, we see a gap of about 0.1 eV. A comparison between Figure 4a and b shows that the band-like features at *U* = 8 eV are much more blurred compared with those at *U* = 6 eV, which can be attributed to the enhanced role of the Coulomb repulsion. Plotting self-energy, depicted in the rightmost panel in Figure 4b, shows that a clear signature of Mott-insulating nature is observed for the $j_{\mathrm{eff}}$ = 1/2 states. Considering the size of the small band gap (∼0.1 eV) in Figure 4b, even at *U* = 8 eV the RuI${}_{3}$ is quite close to the insulator–metal phase boundary. Therefore, we believe that α-RuI${}_{3}$ is likely to be metallic even at the single-layer limit, in contrast to its structural siblings α-RuCl${}_{3}$ and RuBr${}_{3}$.

## 4. Discussion and Summary

It should be commented that the flat-band-like feature observed in the bulk α-RuI${}_{3}$ (see Figure 2b and Figure 3a) is distinct from those reported in kagome lattice systems such as vanadium-based compounds [50]; while the flat bands in kagome lattices originate from the geometric frustration effect, our flat-band-like character in α-RuI${}_{3}$ is from the correlation-induced bandwidth renormalization effect. Comparison between our eDMFT band dispersion (Figure 2b and Figure 3a) and those from DFT + *U* calculations (see, for example, Figure 3 in Ref. [26]) shows the bandwidth renormalization of the $j_{\mathrm{eff}}$ = 1/2 bands from the dynamical electron correlations.

Still, the correlated metallic phase in the α-RuI${}_{3}$, as observed from our results, raises an interesting question: with the presence of an almost vanishing kinetic energy scale due to the flat bands, what would be the role of additional intersite Coulomb repulsion, especially in the potential presence of nontrivial widespread Berry curvature in the momentum space? Such a situation in the absence of spin degree of freedom may lead to exotic phenomena such as fractional Chern insulator phases [51,52]. On the other hand, the flat bands may result in other types of electronic instabilities such as nontrivial charge density waves [53,54] or even superconductivities [55]. Hence, a following study on the topological nature of the metallic $j_{\mathrm{eff}}$ = 1/2 bands in the bulk α-RuI${}_{3}$, especially on the distribution of the Berry phase across the *k*-space and the consequences of including longer-ranged Coulomb interactions, may be necessary in the near future. Another potential study on the effects of tensile epitaxial strain on the single-layer α-RuI${}_{3}$ may also be interesting, since the tensile strain may result in a transition between the trivial and topological band insulating regimes, and also between the band-like and the Mott-insulating regimes.

Overall, we compare the electronic structures of α-RuCl${}_{3}$ and α-RuI${}_{3}$ by employing DFT + DMFT methods. We capture the Mott-insulating nature of α-RuCl${}_{3}$ with the formation of the $j_{\mathrm{eff}}$ = 1/2 local moments. In addition, we report that α-RuI${}_{3}$ is a correlated metal with a correlation-induced flat-band-like feature. Note that this observation can shed light on the puzzling behavior of α-RuI${}_{3}$, especially on its bad-metallic character and sample-dependent magnetic properties, as reported previously [22,23,24]. Our finding suggest that α-RuI${}_{3}$ can be a promising platform for the study of correlated and topological metallic systems.

## Figures and Tables

**Figure 1 nanomaterials-14-00009-f001:**
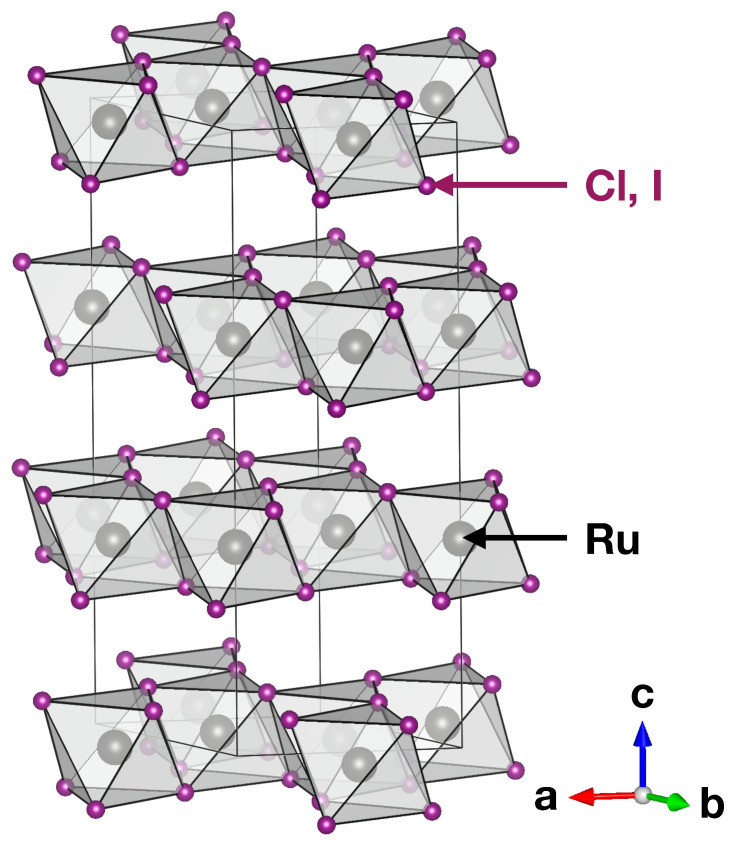
Crystal structure of α-RuCl${}_{3}$ and α-RuI${}_{3}$ with the $R\bar{3}$ space group symmetry.

**Figure 2 nanomaterials-14-00009-f002:**
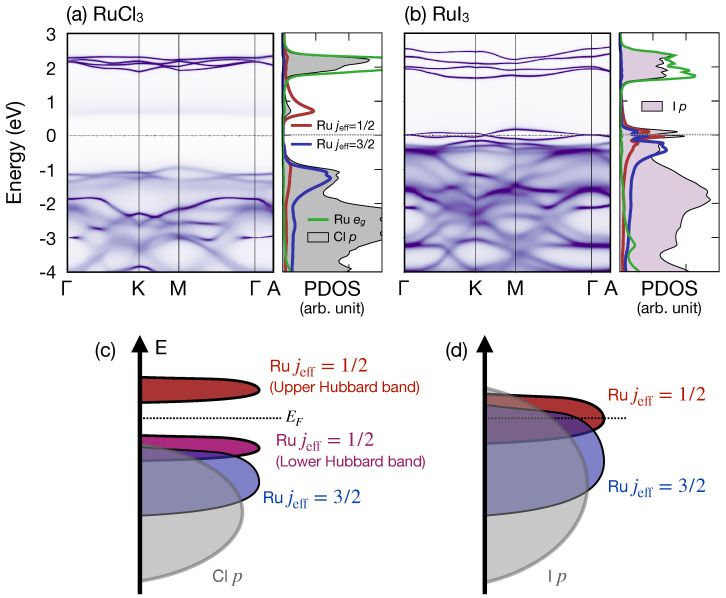
Momentum-dependent and momentum-integrated spectral functions of (**a**) α-RuCl${}_{3}$ and (**b**) α-RuI${}_{3}$, with *U* = 6 eV and $J_{H}$ = 0.8 eV, where orbital-projected spectra are shown on the right panel of each compound. E = 0 is set to be the Fermi level. Schematic energy diagrams for (**c**) α-RuCl${}_{3}$ and (**d**) α-RuI${}_{3}$, where schematic PDOS of Ru $j_{\mathrm{eff}}$ = 1/2, 3/2, and Cl/I *p*-orbitals are depicted in red, blue, and gray, respectively.

**Figure 3 nanomaterials-14-00009-f003:**
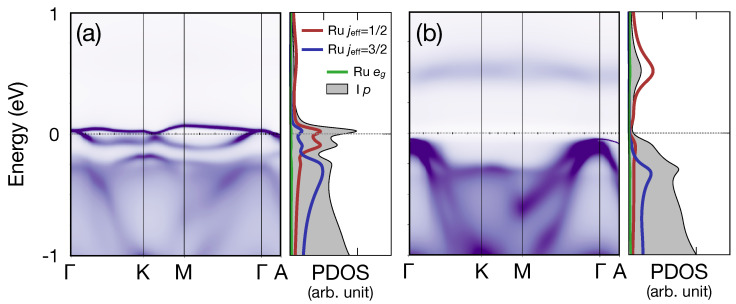
(**a**,**b**) Spectral functions of α-RuI${}_{3}$*U* = 8 and 10 eV, respectively ($J_{H}$ fixed to be 0.8 eV). In the PDOS panels Ru $j_{\mathrm{eff}}$ = 1/2, 3/2, Ru $e_{g}$, and Cl/I *p*-orbital components are depicted in red, blue, green, and gray, respectively.

**Figure 4 nanomaterials-14-00009-f004:**
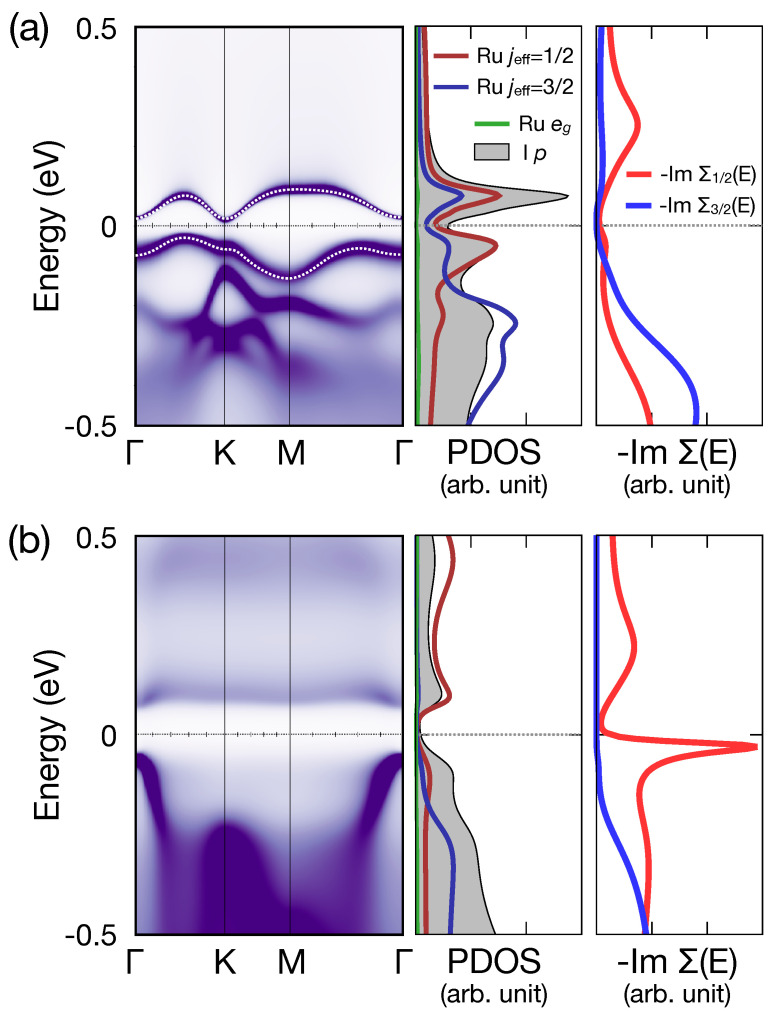
Spectral functions and imaginary part of self-energies of the single-layer α-RuI${}_{3}$ with (**a**) *U* = 6 eV and (**b**) *U* = 8 eV ($J_{H}$ fixed to be 0.8 eV). In the leftmost panel of (**a**), white dotted lines depict quasiparticle band dispersions of the $j_{\mathrm{eff}}$ = 1/2-like bands from a separate spectral function calculation, with the imaginary self-energy set to 0. In the self-energy panels (rightmost panels) red and blue curves depict imaginary part of self-energies (−ImΣ1/2,3/2(E)) for Ru $j_{\mathrm{eff}}$ = 1/2 and 3/2 states, respectively. Note the peak of −ImΣ1/2(E) at the Fermi level when *U* = 8 eV (bright red curve in the rightmost panel of (**b**)), demonstrating the Mott-insulating nature of the $j_{\mathrm{eff}}$ = 1/2 states.

## Data Availability

DMFT code employed in this study can be downloaded from the official webpage (accessed on 12 May 2021) (http://hauleweb.rutgers.edu/tutorials/). The license of the wien2k package can be purchased from the official webpage, (http://www.wien2k.at). All produced data can be directly provided by H.-S.K. (heungsikim@kangwon.ac.kr) upon reasonable request.

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
