# Peer review of "Electronic Structures of Kitaev Magnet Candidates RuCl_3_ and RuI_3"

_nanomaterials, 2023, doi:10.3390/nano14010009_

Round 1

Reviewer 1 Report

Comments and Suggestions for Authors

The authors performed a systematic theoretical investigation on the electronic structures of a-RuCl3 and a-RuI3 by employing DFT+DMFT methods. Obtained results are quite interesting, especially the strong spin-orbit coupling and  quantum spin-Hall insulating phase in a-RuI3 under large U values. This paper can be accepted with the following minor comments.

1) In the caption of Figure 3, there is quotation "(c)". Howere, the corresponding figure is missing. Please carefully check the figure and revise it.

2) The authors discussed about the potential quantum topological phases in a-RuI3 in both the Results and Discussions parts. Please give more concrete evidences or details for this point. 

Comments on the Quality of English Language

Some language and grammar problems should be carefully check and corrected.

Author Response

Dear Reviewer,

We thank your careful consideration, in addition to your positive evaluation and helpful comments on our manuscript. In the revised manuscript we made changes to incorporate criticisms and comments from all of the reviewers, including yours. Please check below for a summary of changes in the revised manuscript, and also point-by-point answers to your comments.

Sincerely yours,

Heung-Sik Kim,

On behalf of Subhasis Samanta and Dukgeun Hong

Summary of Changes:

  1. In the caption of Figure 3 of the previous manuscript, there was a sentence referring to Fig. 3(c), which does not exist. We removed this incorrect reference in the revised manuscript.
  2. We inserted a new paragraph in Page 6 (line 161-171 in the revised manuscript) on the confirmation of the topologically trivial nature of the single-layer RuI3 at U = 6 eV.
  3. In the Computational Methods section of the revised manuscript, new sentences discussing the difference in the choice of the U-values in our eDMFT method compared to conventional DFT+U calculations are inserted.
  4. In the Computational Methods section of the revised manuscript, we added more details on the character of the jeff
  5. At the beginning of the Discussion and Summary section in the revised manuscript, we added a paragraph on the difference between our flat band feature in the bulk RuI3 and those observed in recent kagome compounds.
  6. In the Introduction of the revised manuscript, between line 30 and 53, we inserted sentences clarifying the followings;
    1. Previous discrepancy between experimental and theoretical observations on the nature of RuI3, especially whether the system is metallic or insulating,
    2. How previous DFT+U studies addressed this discrepancy, and
    3. What new aspects our eDMFT results contribute to the community (correlation-induced band flattening and Mott-insulating phenomena) in addition to what’s already discussed by previous DFT+U studies,
    4. How our eDMFT results can provide explanations on the observed bad-metallic character of RuI3.
  7. The last paragraph of the Discussion and Summary section in the revised manuscript was rewritten to emphasize that RuI3is a correlated metal with potential topological phases and instabilities.
  8. Schematic energy diagrams for RuCl3 and RuI3 are inserted as new panels (c) and (d) of Fig. 2 in the revised manuscript.

Responses to the Reviewer’s comments

“The authors performed a systematic theoretical investigation on the electronic structures of a-RuCl3 and a-RuI3 by employing DFT+DMFT methods. Obtained results are quite interesting, especially the strong spin-orbit coupling and quantum spin-Hall insulating phase in a-RuI3 under large U values. This paper can be accepted with the following minor comments.”

We thank the Referee for this positive assessment on our manuscript.

“1) In the caption of Figure 3, there is quotation "(c)". Howere, the corresponding figure is missing. Please carefully check the figure and revise it.”

We apologize for this elementary mistake. In the revised manuscript the reference to the absent Fig. 3(c) is removed in the caption.

“2) The authors discussed about the potential quantum topological phases in a-RuI3 in both the Results and Discussions parts. Please give more concrete evidences or details for this point.”

We thank for this criticism. In the revised manuscript we checked that the band-like-insulator phase of the single-layer RuI3 at U = 6 eV is actually topologically trivial, but can be driven into a quantum spin-Hall phase. We inserted a new paragraph in the Results section (check the second point in the above Summary of Changes).

For the potential outcome of the long-range Coulomb interactions in the correlated metallic phase of the bulk RuI3, and the possible realization of the fractional Chern insulator or other topological phases as commented in the Discussion and Summary section, we would like to mention that rigorous arguments on these claims can be challenging. At this point providing more details on this point beyond we’ve discussed in the section can be difficult, and we would like to ask the Referee’s understanding.

Reviewer 2 Report

Comments and Suggestions for Authors

In this paper, the author investigated the compared electronic structures of a-RuCl3 and a-RuI3 by employing DFT+DMFT method. Since DFT cannot directly deal with strongly correlated interactions in d or f orbitals. MFCS makes some approximations to strict many-body methods and is a good many-body method for describing strongly correlated systems.

The Hubbard parameter U of 6-10eV was selected in this study. Whether the value is too large. The determination of the parameter U should not be empirical, but needs to examine the difference between the DFT and the real potential energy surface, and whether the author can give a more accurate explanation.

Author Response

Dear Reviewer,

We thank your careful consideration, in addition to your positive evaluation and helpful comments on our manuscript. In the revised manuscript we made changes to incorporate criticisms and comments from all of the reviewers, including yours. Please check below for a summary of changes in the revised manuscript, and also point-by-point answers to your comments.

Sincerely yours,

Heung-Sik Kim,

On behalf of Subhasis Samanta and Dukgeun Hong

Summary of Changes:

  1. In the caption of Figure 3 of the previous manuscript, there was a sentence referring to Fig. 3(c), which does not exist. We removed this incorrect reference in the revised manuscript.
  2. We inserted a new paragraph in Page 6 (line 161-171 in the revised manuscript) on the confirmation of the topologically trivial nature of the single-layer RuI3 at U = 6 eV.
  3. In the Computational Methods section of the revised manuscript, new sentences discussing the difference in the choice of the U-values in our eDMFT method compared to conventional DFT+U calculations are inserted.
  4. In the Computational Methods section of the revised manuscript, we added more details on the character of the jeff
  5. At the beginning of the Discussion and Summary section in the revised manuscript, we added a paragraph on the difference between our flat band feature in the bulk RuI3 and those observed in recent kagome compounds.
  6. In the Introduction of the revised manuscript, between line 30 and 53, we inserted sentences clarifying the followings;
    1. Previous discrepancy between experimental and theoretical observations on the nature of RuI3, especially whether the system is metallic or insulating,
    2. How previous DFT+U studies addressed this discrepancy, and
    3. What new aspects our eDMFT results contribute to the community (correlation-induced band flattening and Mott-insulating phenomena) in addition to what’s already discussed by previous DFT+U studies,
    4. How our eDMFT results can provide explanations on the observed bad-metallic character of RuI3.
  7. The last paragraph of the Discussion and Summary section in the revised manuscript was rewritten to emphasize that RuI3is a correlated metal with potential topological phases and instabilities.
  8. Schematic energy diagrams for RuCl3 and RuI3 are inserted as new panels (c) and (d) of Fig. 2 in the revised manuscript.

Responses to the Reviewer’s comments

“In this paper, the author investigated the compared electronic structures of a-RuCl3 and a-RuI3 by employing DFT+DMFT method. Since DFT cannot directly deal with strongly correlated interactions in d or f orbitals. MFCS makes some approximations to strict many-body methods and is a good many-body method for describing strongly correlated systems.”

We thank the Referee for this positive assessment on our results.

“The Hubbard parameter U of 6-10eV was selected in this study. Whether the value is too large. The determination of the parameter U should not be empirical, but needs to examine the difference between the DFT and the real potential energy surface, and whether the author can give a more accurate explanation.”

As the Referee noticed, our value of the on-site Coulomb repulsion U is larger compared to conventional DFT+U calculations of transition metal compounds. But this difference in the optimal value of U originates from the different treatment of screening of Coulomb interactions in our eDMFT method compared to what is done in DFT+U. In short, the U-value used in our eDMFT calculations are more closer to the “bare” (hence larger) U-value, while in DFT+U “screened” (so smaller) U-value should be employed.

Namely, eDMFT includes screening effects into account through higher-order Feynman diagram processes (see the following paper for the long discussion: “Covalency in transition metal oxides within all-electron Dynamical Mean Field Theory”, Phys. Rev. B 90, 075136 (2014), while DFT+U does not. As a result, the effective U that for example, the quasiparticles see within DMFT, is much smaller than the input U in this theory. The nice feature of this approach is that the values of U (and also that of the Hund’s coupling JH) are much more universal and less material dependent than in DFT+U, because most of the screening by the valence electrons is treated within the eDMFT method (while in DFT+U, considerations on the screening process is missing, therefore screened U-values which are smaller than those used in eDMFT calculations should be used).

The choice of the large U-values in the eDMFT method has been tested in a couple of previous benchmarking researches (for example, Haule, Birol, and Kotliar, Phys. Rev. B 90, 075136 (2014) and Mandal et al., npj Comput. Mater. 5, 115 (2019)), where the choice of U=10 eV JH=1 eV across a wide range of 3d transition metal oxides yield reasonable agreements with experimental gap sizes, while in DFT+$U$ fine-tuning of the U-value for each material is necessary to obtain good results (for example, U=6.04, 7.05, 5.91, and 6.88 eV for MnO, NiO, FeO, and CoO, respectively).

In our case, we are employing t2g orbitals of Ru 4d shell as our correlated subspace. Because Ru 4d-orbital is spatially more extended (so leading to smaller U-value), employing smaller U-value compared to 3d transition metal cases is necessary. However, as mentioned above, the U-value used in eDMFT calculations of Ru compounds should be larger than those adopted in DFT+U studies of the same compounds. As mentioned in the manuscript, our choice of U = 6 eV gives good agreement of the band gap size with the experimental value, so we believe that our choice of the U-value is a reasonable one.

To reduce the confusion on our choice of the U-value, we inserted a sentence on addressing this issue in the Computational Methods section. Please check Bullet 3 in the Summary of Changes above.

Reviewer 3 Report

Comments and Suggestions for Authors

It is a nice paper about quantum magnet. I recommand it to be published with a minor revision.

Please give a more detail explaination about Jeff=3/2 and 1/2 of Ru3+, but not Jeff=1/2?

Page 7, line 182-194: for flat band, it is helpful to consider the realtionship between honeycomb a-RuI3 and kagome lattice AV3Sb5(Phys. Rev. Lett., 2020, 125, 247002).

Comments on the Quality of English Language

Page 7, line 171: "Nest" should be "Next".

Page 7, line 181, epitaxial strain on "thesingle-layer" should be "the single layer".

Author Response

Dear Reviewer,

We thank your careful consideration, in addition to your positive evaluation and helpful comments on our manuscript. In the revised manuscript we made changes to incorporate criticisms and comments from all of the reviewers, including yours. Please check below for a summary of changes in the revised manuscript, and also point-by-point answers to your comments.

Sincerely yours,

Heung-Sik Kim,

On behalf of Subhasis Samanta and Dukgeun Hong

Summary of Changes:

  1. In the caption of Figure 3 of the previous manuscript, there was a sentence referring to Fig. 3(c), which does not exist. We removed this incorrect reference in the revised manuscript.
  2. We inserted a new paragraph in Page 6 (line 161-171 in the revised manuscript) on the confirmation of the topologically trivial nature of the single-layer RuI3 at U = 6 eV.
  3. In the Computational Methods section of the revised manuscript, new sentences discussing the difference in the choice of the U-values in our eDMFT method compared to conventional DFT+U calculations are inserted.
  4. In the Computational Methods section of the revised manuscript, we added more details on the character of the jeff
  5. At the beginning of the Discussion and Summary section in the revised manuscript, we added a paragraph on the difference between our flat band feature in the bulk RuI3 and those observed in recent kagome compounds.
  6. In the Introduction of the revised manuscript, between line 30 and 53, we inserted sentences clarifying the followings;
    1. Previous discrepancy between experimental and theoretical observations on the nature of RuI3, especially whether the system is metallic or insulating,
    2. How previous DFT+U studies addressed this discrepancy, and
    3. What new aspects our eDMFT results contribute to the community (correlation-induced band flattening and Mott-insulating phenomena) in addition to what’s already discussed by previous DFT+U studies,
    4. How our eDMFT results can provide explanations on the observed bad-metallic character of RuI3.
  7. The last paragraph of the Discussion and Summary section in the revised manuscript was rewritten to emphasize that RuI3is a correlated metal with potential topological phases and instabilities.
  8. Schematic energy diagrams for RuCl3 and RuI3 are inserted as new panels (c) and (d) of Fig. 2 in the revised manuscript.

Responses to the Reviewer’s comments

“It is a nice paper about quantum magnet. I recommand it to be published with a minor revision.

We thank the Referee for this positive assessment on our results.

“Please give a more detail explaination about Jeff=3/2 and 1/2 of Ru3+, but not Jeff=1/2?”

We are not sure whether we understood this comment correctly, but we guess that the Referee is asking more details on the jefforbitals. Hence we added more details on the nature of the Ru t2g orbitals as the leff=1 manifold and the structure of the jeff orbitals in the Computational Methods section.

“Page 7, line 182-194: for flat band, it is helpful to consider the realtionship between honeycomb a-RuI3 and kagome lattice AV3Sb5(Phys. Rev. Lett., 2020, 125, 247002).”

We thank this valuable comment. In Discussion and Summary section of the revised manuscript, we commented on our flat band feature in the bulk RuI3 in comparison to those observed in kagome systems, as well as citing the reference the Referee mentioned. Please check Bullet 5 in the Summary of Changes above for more details.

“Page 7, line 171: "Nest" should be "Next". Page 7, line 181, epitaxial strain on "thesingle-layer" should be "the single layer".”

Thank you for careful reading. We fixed all of the grammatical issues, including above typos, in our revised manuscript.

Reviewer 4 Report

Comments and Suggestions for Authors

I read the manuscript "Electronic structures of Kitaev magnet candidates RuCl3 and RuI3". The authors study very interesting materials RuCl3 and RuI3, whereas there have already been a number of publications on this topic. Therefore, it is important to emphasize new findings in the manuscript, but the authors failed on this point. I do not recommend publication in this form. Please see the comments below.

1. Sample quality issues in Ref. [26] are not discussed clearly enough. Ref. [26] says, "The reported metallic behavior in RuI3 could have its origin in sample quality." This point should be explicitly written in the introduction part. In the present form, readers will not understand what "discrepancy" is.

2. Finally, the authors say, "Our finding sheds light on...". This is a weak statement. The authors should conclude something about the present discrepancy. Do the authors agree that the reported metallic behavior in RuI3 is due to sample quality?

3. Ref. [27] shows similar theoretical studies on RuCl3 and RuI3. The authors should explicitly write what is new compared to [27]. Ref. [27] uses different values of U and JH, so the authors should discuss this point. The authors explain why they used larger U values.

4. For general readers, simple energy diagrams will be helpful. The diagram, like Fig. 7b in Ref. [27], should be included.

5. (Minor point) Ref. [26] does not have a journal name. Please write.

Author Response

Dear Reviewer,

We thank your careful consideration, in addition to your positive evaluation and helpful comments on our manuscript. In the revised manuscript we made changes to incorporate criticisms and comments from all of the reviewers, including yours. Please check below for a summary of changes in the revised manuscript, and also point-by-point answers to your comments.

Sincerely yours,

Heung-Sik Kim,

On behalf of Subhasis Samanta and Dukgeun Hong

Summary of Changes:

  1. In the caption of Figure 3 of the previous manuscript, there was a sentence referring to Fig. 3(c), which does not exist. We removed this incorrect reference in the revised manuscript.
  2. We inserted a new paragraph in Page 6 (line 161-171 in the revised manuscript) on the confirmation of the topologically trivial nature of the single-layer RuI3 at U = 6 eV.
  3. In the Computational Methods section of the revised manuscript, new sentences discussing the difference in the choice of the U-values in our eDMFT method compared to conventional DFT+U calculations are inserted.
  4. In the Computational Methods section of the revised manuscript, we added more details on the character of the jeff
  5. At the beginning of the Discussion and Summary section in the revised manuscript, we added a paragraph on the difference between our flat band feature in the bulk RuI3 and those observed in recent kagome compounds.
  6. In the Introduction of the revised manuscript, between line 30 and 53, we inserted sentences clarifying the followings;
    1. Previous discrepancy between experimental and theoretical observations on the nature of RuI3, especially whether the system is metallic or insulating,
    2. How previous DFT+U studies addressed this discrepancy, and
    3. What new aspects our eDMFT results contribute to the community (correlation-induced band flattening and Mott-insulating phenomena) in addition to what’s already discussed by previous DFT+U studies,
    4. How our eDMFT results can provide explanations on the observed bad-metallic character of RuI3.
  7. The last paragraph of the Discussion and Summary section in the revised manuscript was rewritten to emphasize that RuI3is a correlated metal with potential topological phases and instabilities.
  8. Schematic energy diagrams for RuCl3 and RuI3 are inserted as new panels (c) and (d) of Fig. 2 in the revised manuscript.

Responses to the Reviewer’s comments

“I read the manuscript "Electronic structures of Kitaev magnet candidates RuCl3 and RuI3". The authors study very interesting materials RuCl3 and RuI3, whereas there have already been a number of publications on this topic. Therefore, it is important to emphasize new findings in the manuscript, but the authors failed on this point. I do not recommend publication in this form. Please see the comments below.”

We thank the Referee for this critical assessment on our manuscript. As the Referee pointed out, there are several first-principles studies on RuCl3 and RuI3. However all of previous first-principles studies have employed density functional theory, and our result is the first dynamical mean-field theory study on the electronic structure of RuCl3 and RuI3. It is true that our conclusion is somewhat similar to that presented in Ref. 27 (Liu et al., Phys. Rev. B 107, 165134 (2023)) of the previous manuscript, but whether DFT+U and DFT+DMFT, which are completely different methodologies, would yield qualitatively similar results still remain a valid question. This question is resolved by our manuscript. Also, a direct confirmation of the Mott-insulating phase in RuCl3 and the bad-metallic character in RuI3 due to the correlation-induced band flattening are another new results of ours in comparison to Liu et al.

In response to the Reviewer’s criticism, In the revised manuscript we emphasized the difference between Liu et al. and our results. Please check Bullet 6 in the Summary of Changes above for more details.

“1. Sample quality issues in Ref. [26] are not discussed clearly enough. Ref. [26] says, "The reported metallic behavior in RuI3 could have its origin in sample quality." This point should be explicitly written in the introduction part. In the present form, readers will not understand what "discrepancy" is.”

We appreciate this valuable comment. Ref. 26 claims that the bulk pristine RuI3 should be insulating based on their first-principles DFT+U calculations, while experimentally (as reported in Ref. 23 and Ref. 25) RuI3 is reported to be metallic. Ref. 26 it is speculated that the observed bad-metallic nature of the RuI3 may originate from the sample quality issue, specifically due to metallic grain boundaries between insulating RuI3 grains. Later DFT+U studies, including Ref. 27 in the previous manuscript, provide a partial explanation of this ‘discrepancy’ between experimental and theoretical observations by choosing a suitable U-value that yields insulating and metallic phases in RuCl3 and RuI3, respectively. However, the exceptionally bad-metallic character as reported in Ref. 25, may not be able to be explained from the metallic band structures as obtained from DFT. By “discrepancy” we meant these issues, on which we were not very clear in the previous manuscript.

In the Introduction part of the revised manuscript we explicitly mention this history and what the ‘discrepancy’ means. Please check Bullet 6 in the Summary of Changes above for more details.

“2. Finally, the authors say, "Our finding sheds light on...". This is a weak statement. The authors should conclude something about the present discrepancy. Do the authors agree that the reported metallic behavior in RuI3 is due to sample quality?”

We believe that, from our eDMFT results, the bulk RuI3 is a correlated metal, and that the observed bad-metallic character is an outcome of dynamic electron correlations and the resulting strong bandwidth renormalization. To make this point clear, we revised the last paragraph of the Discussion and Summary section. Please check Bullet 7 in the Summary of Changes above.

.

“3. Ref. [27] shows similar theoretical studies on RuCl3 and RuI3. The authors should explicitly write what is new compared to [27]. Ref. [27] uses different values of U and JH, so the authors should discuss this point. The authors explain why they used larger U values."

We thank the Referee for this valuable comment. In the Introduction of the revised manuscript, we outline the difference between our results and previous DFT+U ones (for example, Ref. 27). Please check Bullet 6 in the Summary of Changes above for more details.

On our choice of U, as the Referee noticed, our value of the on-site Coulomb repulsion U is larger compared to conventional DFT+Ucalculations of transition metal compounds. But this difference in the optimal value of U originates from the different treatment of screening of Coulomb interactions in our eDMFT method compared to what is done in DFT+U. In short, the U-value used in our eDMFT calculations are more closer to the “bare” (hence larger) U-value, while in DFT+U “screened” (so smaller) U-value should be employed.

Namely, eDMFT includes screening effects into account through higher-order Feynman diagram processes (see the following paper for the long discussion: “Covalency in transition metal oxides within all-electron Dynamical Mean Field Theory”, Phys. Rev. B 90, 075136 (2014), while DFT+U does not. As a result, the effective U that for example, the quasiparticles see within DMFT, is much smaller than the input U in this theory. The nice feature of this approach is that the values of U (and also that of the Hund’s coupling JH) are much more universal and less material dependent than in DFT+U, because most of the screening by the valence electrons is treated within the eDMFT method (while in DFT+U, considerations on the screening process is missing, therefore screened U-values which are smaller than those used in eDMFT calculations should be used).

The choice of the large U-values in the eDMFT method has been tested in a couple of previous benchmarking researches (for example, Haule, Birol, and Kotliar, Phys. Rev. B 90, 075136 (2014) and Mandal et al., npj Comput. Mater. 5, 115 (2019)), where the choice of U=10 eV JH=1 eV across a wide range of 3d transition metal oxides yield reasonable agreements with experimental gap sizes, while in DFT+$U$ fine-tuning of the U-value for each material is necessary to obtain good results (for example, U=6.04, 7.05, 5.91, and 6.88 eV for MnO, NiO, FeO, and CoO, respectively).

In our case, we are employing t2g orbitals of Ru 4d shell as our correlated subspace. Because Ru 4d-orbital is spatially more extended (so leading to smaller U-value), employing smaller U-value compared to 3d transition metal cases is necessary. However, as mentioned above, the U-value used in eDMFT calculations of Ru compounds should be larger than those adopted in DFT+U studies of the same compounds. As mentioned in the manuscript, our choice of U = 6 eV gives good agreement of the band gap size with the experimental value, so we believe that our choice of the U-value is a reasonable one.

To reduce the confusion on our choice of the U-value, we inserted a sentence on addressing this issue in the Computational Methods section. Please check Bullet 3 in the Summary of Changes above.

“4. For general readers, simple energy diagrams will be helpful. The diagram, like Fig. 7b in Ref. [27], should be included.”

We thank for this great suggestion. In the new Fig. 2, schematic energy diagrams for RuCl3 and RuI3 are inserted as new panels (c) and (d).

“5. (Minor point) Ref. [26] does not have a journal name. Please write.”

We apologize for this elementary mistake, and thank the Referee for this careful reading. This was fixed in the revised manuscript.

Round 2

Reviewer 4 Report

Comments and Suggestions for Authors

I read the revised manuscript "Electronic structures of Kitaev magnet candidates RuCl3 and RuI3". I found improvement, especially in the summary part: "In addition, we report that α-RuI3 is a correlated metal with a correlation-induced flat-band-like feature." Now I recommend publication in this form.